# The climate effects of increasing ocean albedo: An idealized representation of solar geoengineering

Ben Kravitz[1], Philip J. Rasch[1], Hailong Wang[1], Alan Robock[2], Corey Gabriel[3], Olivier Boucher[4], Jason N. S. Cole[5], Jim Haywood[6,7], Duoying Ji[8], Andy Jones[6], Andrew Lenton[9], John C. Moore[8], Helene Muri[10,11], Ulrike Niemeier[12], Steven Phipps[13,14], Hauke Schmidt[12], Shingo Watanabe[15], Shuting Yang[16], and Jin-Ho Yoon[17]

[1]Atmospheric Sciences and Global Change Division, Pacific Northwest National Laboratory, Richland, WA, USA
[2]Department of Environmental Sciences, Rutgers University, New Brunswick, NJ, USA
[3]Scripps Institution of Oceanography, La Jolla, CA, USA
[4]Laboratoire de Météorologie Dynamique, CNRS / Sorbonne Université, Paris, France
[5]Environment and Climate Change Canada, Toronto, Canada
[6]Met Office Hadley Centre, Exeter, UK
[7]College of Engineering, Mathematics, and Physical Sciences, University of Exeter, Exeter, UK
[8]State Key Laboratory of Earth Surface Processes and Resource Ecology, College of Global Change and Earth System Science, Beijing Normal University, Beijing, China
[9]CSIRO Oceans and Atmosphere, Hobart, Tasmania, Australia
[10]Department of Geosciences, University of Oslo, Oslo, Norway
[11]Department of Energy and Process Engineering, Norwegian University of Science and Technology, Trondheim, Norway
[12]Max Planck Institute for Meteorology, Hamburg, Germany
[13]Climate Change Research Centre, University of New South Wales, Sydney, Australia
[14]Institute for Marine and Antarctic Studies, University of Tasmania, Hobart, Tasmania, Australia
[15]Japan Agency for Marine-Earth Science and Technology, Yokohama, Japan
[16]Danish Meteorological Institute, Copenhagen, Denmark
[17] School of Earth Sciences and Environmental Engineering, Gwangju Institute of Science and Technology, Gwangju, South Korea

**Correspondence:** Ben Kravitz, P.O. Box 999, MSIN K9-30, Richland, WA 99352, USA. (ben.kravitz@pnnl.gov)

**Abstract.** Geoengineering, or climate intervention, describes methods of deliberately altering the climate system to offset anthropogenic climate change. As an idealized representation of near-surface solar geoengineering over the ocean, such as marine cloud brightening, this paper discusses experiment G1ocean-albedo of the Geoengineering Model Intercomparison Project (GeoMIP), involving an abrupt quadrupling of the $CO_2$ concentration and an instantaneous increase in ocean albedo

5    to maintain approximate net top-of-atmosphere radiative flux balance. Eleven Earth System Models are relatively consistent in their temperature, radiative flux, and hydrological cycle responses to this experiment. Due to the imposed forcing, air over the land surface warms by a model average of 1.14 K, while air over most of the ocean cools. Some parts of the near-surface air temperature over ocean warm due to heat transport from land to ocean. These changes generally resolve within a few years, indicating that changes in ocean heat content play at most a small role in the warming over the oceans. The hydrological

10    cycle response is a general slowing down, with high heterogeneity in the response, particularly in the tropics. While idealized, these results have important implications for marine cloud brightening, or other methods of geoengineering involving spatially

heterogeneous forcing, or other general forcings with a strong land/ocean contrast. It also reinforces previous findings that keeping top-of-atmosphere net radiative flux constant is not sufficient for preventing changes in global mean temperature.

*Copyright statement.* TEXT

## 1 Introduction

Geoengineering (also called "climate intervention") describes a set of technological approaches to reduce the effects of climate change by deliberately intervening in the climate system (e.g., Shepherd et al., 2009). There are two broad categories of geoengineering that are commonly discussed: solar geoengineering (modifying the amount of shortwave radiation incident at the surface; NAS, 2015b) and carbon dioxide removal (NAS, 2015a). There are also proposals, such as cirrus cloud thinning (Mitchell and Finnegan, 2009) that do not fit neatly into either of these two categories. In all subsequent discussions in this manuscript, we only discuss solar geoengineering methods.

Two of the most commonly proposed methods of global geoengineering are stratospheric sulfate aerosol geoengineering and marine cloud brightening (MCB). Comparison of the different climate effects of these two methods (e.g., Niemeier et al., 2013; Crook et al., 2015) reveals that, among other things, the spatial distribution of the applied forcing strongly affects the climate effects. Many of the effects of sulfate geoengineering can be reasonably well approximated by a uniform reduction in shortwave radiative flux reaching the surface (Kalidindi et al., 2014). Conversely, MCB targets low clouds over oceans (Latham, 1990), which are not ubiquitous. In addition, there are higher-order effects due to the altitude at which the shortwave scattering occurs, including multiple scattering effects, infrared absorption of shortwave and longwave radiative flux by sulfate aerosols or cloud particles, and absorption of shortwave radiative flux by atmospheric $CO_2$ and water vapor (e.g., Kravitz et al., 2013b).

Idealized simulations of solar geoengineering are useful in the context of multi-model intercomparisons, in that they capture many of the effects of more complicated methods of representing geoengineering, yet can be performed by a wide variety of models. In simulations conducted under the Geoengineering Model Intercomparison Project (GeoMIP; Kravitz et al., 2011), an idealized method of representing stratospheric sulfate aerosol geoengineering is via reductions in total solar irradiance. As an example of this representation, experiment G1 involved offsetting the global radiative flux imbalance from a quadrupling of the $CO_2$ concentration via solar reduction. Thus far, 15 models have participated in this simulation, providing information about model commonalities and differences in the global climate response, including effects on temperature, the hydrological cycle, cryosphere, terrestrial biosphere, and extreme events (Schmidt et al., 2012; Kravitz et al., 2013a, b; Tilmes et al., 2013; Moore et al., 2014; Glienke et al., 2015; Curry et al., 2014, among numerous other studies). The GeoMIP website (http://climate.envsci.rutgers.edu/GeoMIP/) provides an up-to-date list of publications using GeoMIP model output.

While total solar irradiance reductions are straightforward to simulate in all models, this idealization is not a good approximation of MCB, nor of near-surface solar geoengineering approaches over the ocean in general. The dominant effect of MCB

would be an increase in albedo of marine low clouds through aerosol effects. More generally, changes in the albedo near the marine surface (such as in the G4Foam experiment; Gabriel et al., 2017) can produce different signatures from reductions in energy input at the top of the atmosphere, particularly in terms of spatial distribution. While some forms of albedo modification like stratospheric sulfate aerosol geoengineering operate over broad areas (on a hemispheric or larger scale), albedo changes produced by near-surface marine geoengineering would likely operate on smaller spatial scales and be concentrated over particular oceanic regions.

In this study, we investigate the climate effects of using ocean albedo increases to offset $CO_2$ warming and compare those effects with those of total solar irradiance reduction (experiments are described in more detail in the following section). All simulations were conducted under the auspices of GeoMIP, allowing us to characterize a range of model responses to these different idealized methods of representing solar geoengineering.

## 2   Methodology and Description

Our analyses focus on four simulations: (1) a preindustrial control simulation (piControl), (2) a simulation in which the $CO_2$ concentration is abruptly quadrupled from its preindustrial value (abrupt4xCO2), (3) a simulation in which the net radiative flux imbalance in abrupt4xCO2 is offset by a reduction in total solar irradiance (G1), and (4) a simulation in which the net top-of-atmosphere radiative flux imbalance in abrupt4xCO2 is offset by an increase in ocean albedo everywhere by a uniform factor (G1ocean-albedo). piControl and abrupt4xCO2 are standard experiments in the Coupled Model Intercomparison Project Phase 5 (CMIP5; Taylor et al., 2012). G1 is described further by Kravitz et al. (2011), and many of the gross features of the results are described by Kravitz et al. (2013a). All models participating in experiment G1 needed to reduce model total solar irradiance by 3.5-5.0% to offset the radiative forcing from a quadrupling of the $CO_2$ concentration. In G1ocean-albedo, the ocean surface albedo was increased abruptly at the start of the simulation such that net top-of-atmosphere radiative flux perturbation was within $\pm 0.1$ W m$^{-2}$ of the piControl value in an average over years 21-30 of simulation. Based on preliminary simulations described by Kravitz et al. (2013c), it took approximately 20 years for the climate to reach steady state after an abrupt simultaneous change in the $CO_2$ concentration and the ocean albedo. As will be shown in subsequent sections, once the appropriate value of ocean albedo increase is found and imposed, the climate system adjusts rapidly, requiring at most a few years to reach a steady state in global mean temperature (as was the case in experiment G1; Kravitz et al., 2013a). Table 1 lists the models participating in this study, including relevant references and the required change in albedo to meet the objectives of experiment G1ocean-albedo. A similar table for experiment G1 is given by Kravitz et al. (2013a). One of the advantages of G1ocean-albedo is that, like G1, all models can conduct this simulation fairly easily. Supplemental Table S1 quantifies how well each model achieved radiative balance in the G1 and G1ocean-albedo experiments.

Supplemental Table S2 quantifies temperature trends in each participating model over years 11-50 of simulation. The mean model trend over this period is approximately zero K decade$^{-1}$ (to four decimal places), and with little exception, the trends in G1 and G1ocean-albedo are an order of magnitude smaller than the trends in the abrupt4xCO2 simulation. As such, for the purpose of analysis, we assume that "slow responses," i.e., responses operating on time scales longer than a few years

(e.g., Andrews and Forster, 2010; Sherwood et al., 2015) are negligible in the G1 and G1ocean-albedo simulations. We do not separate results into rapid adjustment and slow response timescales, and with the exception of time series plots, all figures show averages over the years 11-50 of simulation, which we take as a sufficient indication of the dominant climate response after the transient response has resolved.

Except where indicated, all plots show the mean model response. All values in the text are reported as mean (min to max), where mean indicates the all-model mean for that particular quantity, min is the lower bound of the range of model responses, and max is the upper bound of the range of model responses. In all maps, stippling indicates where fewer than 75% of the models agree on the sign of the response. All models in Table 1 were able to provide output for all variables except for cloud radiative forcing. The models included in cloud forcing analyses are BNU-ESM, CanESM2, CESM-CAM5.1-FV, HadGEM2-
ES, IPSL-CM5A-LR, and MPI-ESM-LR. Supplemental Tables S1-S15 provide more quantitative information for all of the analyses presented in this study.

## 3 Results

### 3.1 Albedo and Temperature

Figure 1 shows the change in albedo at the top-of-atmosphere and at the surface for the abrupt4xCO2, G1, and G1ocean-albedo
simulations, where albedo is defined as the ratio of upwelling to downwelling all-sky shortwave radiative flux. Quantitative values are given in Supplemental Tables S3 and S4. Results for abrupt4xCO2 and G1 are consistent with known responses of an increase in absorbed shortwave by increased $CO_2$, reduced cloud cover, and reduced snow and sea ice cover (e.g., Schmidt et al., 2012; Kravitz et al., 2013b). These result in a broad decrease in albedo at the top of atmosphere and a decrease in surface albedo in many regions with substantial snow and ice cover. G1ocean-albedo retains many of these local high latitude features,
but with large albedo increases over ocean, consistent with the experimental design and imposed forcing.

    Figures 2 and 3 expand upon this picture by showing changes in shortwave and longwave cloud forcing and clear sky flux in G1 and G1ocean-albedo. In Figure 2, cloud forcing is defined as all-sky minus clear-sky radiative flux measured at the top of the atmosphere. Positive shortwave values and negative longwave values in Figure 2 are indicative of less cloud cover. In Figure 3, values indicate changes in top-of-atmosphere net clear sky flux, where net is defined as downward minus upward.
Positive values indicate less upward flux in the perturbed experiment (G1 or G1ocean-albedo), and negative values indicate more upward flux.

    Kravitz et al. (2013b) showed that cloud cover in G1 tends to be reduced, which is consistent with what is depicted in Figure 2 over broad swaths of the globe. For G1ocean-albedo, cloud cover is reduced over most ocean regions and large portions of land. Exceptions include negative shortwave and positive longwave values over the Arctic, much of Africa, South Asia, Australia,
and the leeward side of the Andes. The results of Figure 3 are consistent with an increase in the $CO_2$ concentration, with more absorption of shortwave and more outgoing longwave radiative flux. Exceptions are many of the same regions as in Figure 2, which show negative (or less positive) shortwave values and less negative longwave values. Thus, over most regions of the globe, the results are consistent with a combination of increased $CO_2$ and less cloud cover. Over the other regions (named

previously), Figure 2 would indicate that cloud cover increases, which would result in less shortwave absorption and less outgoing longwave radiative flux, consistent with the results in Figure 3. These changes in cloudiness have implications for the hydrologic cycle, which we revisit in Section 3.5.

Figures 2 and 3 admittedly only report the first-order explanations of the radiative flux changes in G1 and G1ocean-albedo. Second-order effects could include additional shortwave absorption by clouds or feedbacks on water vapor flux due to reduced evaporation. Additional work is needed to better understand the role of individual flux changes and processes on clouds and circulation patterns.

Figure 4 shows changes in global mean, land mean, and ocean mean surface air temperature for the G1 and G1ocean-albedo multi-model ensembles. Quantitative values are provided in Supplemental Table S5. Whereas the G1 simulation largely offsets global temperature changes due to increased $CO_2$ concentration, G1ocean-albedo is approximately 0.36 K (-0.12 to 1.20) warmer than the control simulation. This is predominantly due to warming over land by 1.14 K (0.41 to 1.83). The temperature results in Figure 4 indicate that the temperature change happens within approximately the first year, and while some models show a slight trend in temperature over the 50-year G1ocean-albedo simulation (Supplemental Table S2), in general, any such trends are small, especially as compared to the warming in the abrupt4xCO2 simulation. This lack of substantial transient behavior after an initial fast response indicates that G1ocean-albedo has entered a new approximate steady state.

Figure 5 shows spatial patterns of change in temperature and top-of-atmosphere net radiative flux. (Also see Supplemental Tables S5 and S6.) The temperature changes are broadly consistent with the net radiative flux changes in the respective experiments. As was discussed by Kravitz et al. (2013a), G1 results in an "overcooling" of the tropics and an "undercooling" of the poles, consistent with offsetting the ubiquitous longwave forcing from $CO_2$ with a latitudinally dependent reduction in shortwave. G1ocean-albedo shows warming at high latitudes, over land regions, and in some ocean regions near or downwind of large continents, with the remaining ocean regions generally showing cooling. This warming pattern downwind of large continents does not have a seasonal component, although some individual models show more warming than others (not shown).

While the warming over land is easily explainable from first principles, the temperature response over the ocean is heterogeneous (likely due to clouds – see above), and it is perhaps somewhat counterintuitive that on average temperatures over the global oceans do not decrease. Because net top-of-atmosphere radiative flux is approximately zero in G1ocean-albedo, the global warming cannot be the result of energy being added to or subtracted from the climate system, and instead must be the result of energy redistribution. Three hypotheses for why these temperature change patterns look the way they do (which will be tested in subsequent sections) include

1. Based on energy balance arguments, G1ocean-albedo should experience global average warming.

2. Most warming over oceanic regions is due to transport of heat from land to ocean.

3. Any contributions to temperature or radiative flux changes from changes in ocean heat content are small on the timescales being evaluated here.

### 3.2 Hypothesis 1: Energy Balance

The Earth system can be considered as a simple surface-atmosphere energy budget model

$$\frac{S(1-A)}{4} = (1 - \epsilon/2)\sigma T_s^4 \tag{1}$$

where $S$ is total solar irradiance at the top of the atmosphere (i.e., the solar "constant"), $A$ is albedo of Earth, $\epsilon$ is the longwave emissivity of the atmosphere, and $T_s$ is surface temperature. In this model, $T_s = 2^{1/4} T_a$, where $T_a$ is atmospheric temperature.

Taking the total differential yields

$$\frac{dS(1-A)}{4} - \frac{SdA}{4} = \left(1 - \frac{\epsilon}{2}\right) 4\sigma T_s^3 dT_s - \frac{d\epsilon}{2}\sigma T_s^4 \tag{2}$$

Isolating $dT_s$ yields

$$dT_s = \left[\frac{dS(1-A)}{4} - \frac{SdA}{4} + \frac{d\epsilon}{2}\sigma T_s^4\right] / \left[\left(1 - \frac{\epsilon}{2}\right) 4\sigma T_s^3\right] \tag{3}$$

Simplifying,

$$dT_s = \frac{dS(1-A)}{16\sigma T_s^3(1-\epsilon/2)} - \frac{SdA}{16\sigma T_s^3(1-\epsilon/2)} + \frac{d\epsilon/2}{1-\epsilon/2}\frac{T_s}{4} \tag{4}$$

From Equation 1 and using $T_s = 286.491$ K (the average piControl value from the Earth System Models), $S = 1366$ W m$^{-2}$, and $A = 0.3$, it follows that $\epsilon = 0.748$.

Equation 4 can be augmented to consider changes in land and ocean components separately. $F_\ell$ and $F_o$ are the land and ocean fractions, 0.3 and 0.7 respectively, such that $F_\ell + F_o = 1$. The land and ocean albedos are $A_\ell$ and $A_o$, respectively. Greenhouse gases are assumed to be well mixed (i.e., $d\epsilon$ is the same over land and ocean). Then after solving for change in ocean temperature $dT_{s,o}$, Equation 4 becomes

$$dT_{s,o} = \frac{1}{F_o}\left[\frac{(F_\ell dS_\ell + F_o dS_o)(1-A)}{16\sigma T_s^3(1-\epsilon/2)} - \frac{S(F_\ell dA_\ell + F_o dA_o)}{16\sigma T_s^3(1-\epsilon/2)} + \frac{d\epsilon/2}{1-\epsilon/2}\frac{T_s}{4} - F_\ell dT_{s,\ell}\right] \tag{5}$$

Nearly all of the variables on the right sides of Equations 4 and 5 can be solved from values provided in the supplemental material, values provided above, and $dS/S = -0.042$ (Kravitz et al., 2013b). The only variable that is difficult to solve for in this idealized context is $d\epsilon$, representing changes in emissivity. Such changes can occur due to changes in the $CO_2$ concentration (or other greenhouse gases), changes in water vapor, or changes in cloud cover. Estimating this quantity using the abrupt4xCO2 scenario would correctly capture changes in emissivity due to $CO_2$ changes under the G1ocean-albedo simulation, but it would likely overestimate contributions due to water vapor because of tropospheric warming. As such, estimates of $d\epsilon$ under G1ocean-albedo will be calculated using G1, which will capture changes in emissivity from the $CO_2$ changes but without large changes in atmospheric water vapor. Admittedly, water vapor and cloud cover will likely differ between G1 and G1ocean-albedo, rendering this estimate imperfect. However, we think this process yields a more appropriate result than using abrupt4xCO2.

Using Equation 4 and substituting $dT_s = 0$ K, $dS = 1366 \cdot (-0.042)$ W m$^{-2}$, $A = 0.3$, $T_s = 286.491$ K, $\epsilon = 0.748$, and $dA = -0.007$ (Supplemental Table S3) yields $d\epsilon = 0.0401$. For G1, each of the three terms on the right side of Equation 4 are

then -3.01, 0.72, and 2.29 K, respectively. The first of these terms corresponds to solar changes, the second term is for planetary albedo changes, and the third term is for emissivity (greenhouse gas) changes.

From the Supplemental Tables, for G1ocean-albedo, $dA_o = 0.023$, $dA_\ell = -0.004$, $dS_\ell = dS_o = 0$, and $dT_{s,\ell} = 1.14$ K. Then substituting into Equation 5 yields $dT_{s,o} = 0.61$ K, which is higher than the Earth System Model ensemble average of

0.03 K. For comparison with the values from G1, the term corresponding to changes in solar input is 0 K, the term corresponding to changes in albedo is -1.52 K, and the term for changes in emissivity is 2.29 K. By Equation 4, these values yield a global mean temperature change of 0.77 K, which is higher than the Earth System Model ensemble average of 0.36 K.

This simple energy balance formulation clearly cannot incorporate all of the feedbacks and complex behaviors of the Earth System Models. Nevertheless, based on energy balance constraints, G1ocean-albedo results in both land and ocean warming.

However, the values recovered by the energy balance model are not consistent with the results of the Earth System Models for G1ocean-albedo. To account for these differences, we turn to circulation changes, which are described in the following section.

### 3.3 Hypothesis 2: The role of Land-Ocean Energy Transport (LOET)

Although the air over the ocean warms somewhat in G1ocean-albedo, it does not warm uniformly. Figure 5 shows that much of the warming over the ocean is in areas near land, indicating the potential for some of the heating energy over land to be

transported to ocean regions. Indeed, the oceans far from land experience cooling, which is consistent with expectations for a large increase in albedo (Table 1).

Transport of heating energy from land to ocean can be quantified via calculating what Geoffroy et al. (2015) call horizontal energy transport, and which we call land-ocean energy transport (LOET), as it represents an aggregate transport of energy from the atmosphere over the land (averaged over all land regions) to the atmosphere over the ocean (averaged over all ocean

regions). Geoffroy et al. (2015) provide a more detailed description, calculation, and validation of this concept using a three-box energy balance model that can be fitted to changes in land/ocean temperature and TOA energy imbalance such that the model captures the relevant energy transport dynamics; we repeat here only the calculations germane to our discussions.

Gregory et al. (2004) describe a method of estimating adjusted radiative forcing and the aggregate strength of global feedbacks via linear regression of the net global, annual mean TOA radiative flux imbalance ($\Delta R$) against the global, annual mean

temperature change ($\Delta T$) in response to a forcing. The y-intercept of the regression line gives an estimate of adjusted radiative forcing ($\mathcal{F}$), and the negative of the slope of the regression line gives the feedback parameter ($\lambda$). Similarly, one can perform regression just over land-averaged quantities (denoted with the subscript $\ell$) or just over ocean quantities (subscript $o$). Feedback parameter values are provided in Table 2.

In addition, as is derived in detail by Geoffroy et al. (2015), one can regress $\Delta T_\ell$ against $\Delta T_o$ to obtain the equation

$$\Delta T_\ell = \frac{\alpha_o/F_\ell}{\lambda_\ell + \alpha_\ell/F_\ell}\delta T_o + \frac{\mathcal{F}}{\lambda_\ell + \alpha_\ell/F_\ell} \tag{6}$$

where $\alpha_\ell$ is the land heat transport parameter (units of W m$^{-2}$ K$^{-1}$), $\alpha_o$ is the ocean heat transport parameter, and $F_\ell$ is the land fraction (approximately 0.3). If one solves this equation for $\alpha_\ell$ and $\alpha_o$, then one can define

$$\Delta Q = \alpha_\ell \Delta T_\ell - \alpha_o \Delta T_o \tag{7}$$

The quantity $\Delta Q$ is the time-dependent LOET (units of W m$^{-2}$).

Figure 6 provides calculations of LOET for the simulations presented here. See Supplemental Table S7 for more details on individual model values. In the abrupt4xCO2 simulation, changes in LOET are positive with respect to piControl (indicating an increase in heat transport from the land to the ocean) and decrease in magnitude steadily over the course of the simulation; these results are discussed in more detail by Geoffroy et al. (2015).

In experiment G1, LOET increases by a model-dependent constant value and remains relatively unchanged over the course of the simulation. Although the air temperature over land in G1 increases slightly, and the air temperature over ocean decreases slightly (Kravitz et al., 2013a), the temperature changes in G1 are more latitude-dependent than representative of a clear land-ocean contrast (Figure 5), so it is perhaps not unexpected that LOET would be small.

Experiment G1ocean-albedo exhibits a strong land-ocean contrast in temperature (Figure 5), and the response is in steady state after a few years. As such, consistent with the behavior of other fluxes, LOET in G1ocean-albedo does not show transient behavior. LOET in G1ocean-albedo is approximately 2.20 (1.35 to 3.21) W m$^{-2}$, which is larger than in the other experiments examined here.

### 3.4 Hypothesis 3: Atmospheric Column Energetics and Net Energy Flux into the Oceans

An additional potential source of energy to the atmosphere is a reduction in net ocean heat uptake. Calculating changes in ocean heat uptake are challenging and not particularly revealing in this study for three reasons:

1. It is possible that the models used in simulating G1ocean-albedo were not entirely spun up to steady state. As such, any remaining imbalances could manifest as changes in ocean heat content. In principle, one could subtract off the preindustrial control value, which likely has a similar trend in ocean heat content arising from spinup. However, this would not remove the influence of nonlinearities (state dependence), so there is no way to guarantee that the signal is entirely due to the G1ocean-albedo forcing.

2. As is seen in Supplemental Table S1, not all models were able to achieve top-of-atmosphere net radiative flux balance over the course of the simulation. These small changes can lead to large changes in ocean heat content over the course of a 50-year simulation, consistent with CMIP5 models (Hobbs et al., 2016). For example, a 0.1 W m$^{-2}$ imbalance over a 50-year period can lead to an additional $5.5 \times 10^{22}$ J of energy incident at the ocean surface. As such, we are unable to properly assess the degree to which ocean heat content changes may be due to small imbalances.

3. Ocean heat content can be (and is often) calculated up to a certain depth, meaning calculations of it can be sensitive to redistribution of heat to/from lower depths, obscuring the signal of the forcing.

As an alternative, we calculate net energy exchange across the surface in terms of changes in radiative and turbulent fluxes. Kravitz et al. (2013b) calculated energetics changes in the entire atmospheric column. However, because we are only interested in net surface fluxes, we calculate

$$\Delta B = \Delta R_{\text{surf}} + \Delta SH + \Delta LH \tag{8}$$

where $\Delta R_{\mathrm{surf}}$ is the change in net surface radiative flux (shortwave and longwave), $\Delta SH$ is change in sensible heat flux from the atmosphere to the surface, and $\Delta LH$ is change in latent heat flux from the atmosphere to the surface. By convention, all fluxes are positive downward unless specifically noted. Calculations of individual terms in this budget, as well as of $\Delta B$, are provided in Supplemental Tables S8-S12. Because these calculations are performed at the surface, no advection term (e.g.,
LOET) is needed, and $\Delta B$ is well defined as a land or ocean average.

    Figure 7 shows the all-model mean for all of the terms in Equation 8. Several clear conclusions emerge. The first is that $\Delta B$ is approximately zero globally, over land, and over ocean for nearly the entire 50-year period, after an initial rapid adjustment that resolves within a few years. With the exception of latent heat over land, all fluxes for G1ocean-albedo reach a steady state after a few years (Figure 7), and even latent heat flux over land reaches an approximate steady state within ten years. If
$\Delta B$ indeed serves as a useful proxy for global net energy flux into or out of the ocean, then these results indicate that there is no sizable contribution to atmospheric energetics by changes in global mean ocean heat content. Moreover, even if $\Delta B$ were not zero over ocean, global mean ocean heat content changes would still be an insufficient explanation for global mean temperature changes due to incongruent timescales. The oceanic mixed layer operates on an approximately decadal timescale, but all transient behavior in these simulations is resolved well before ten years. The transient response is much more consistent
with a land surface time scale, which is on the order of 1-3 years. As such, it seems plausible that the temperature changes over ocean in G1ocean-albedo are due to land processes and land surface feedbacks rather than ocean heat content changes. This is not to say that the ocean plays no role in the observed temperature changes. Rather, given the discussions in this section and the two previous sections, the role of global mean ocean heat content in causing temperature changes over the ocean in G1ocean-albedo (over the timescales being analyzed here) is likely small. Because forcings and feedbacks are likely to be realized
heterogeneously, there may be roles for local changes or for changes in patterns of circulation (e.g., the Atlantic meridional overturning circulation) in altering oceanic heat content. However, such analyses are beyond the scope of the present work.

    The remainder of the results in Figure 7 are consistent with the applied forcing. There is a large sensible heat flux increase from the land to the atmosphere of 2.87 (-0.99 to 6.00) W m$^{-2}$, with a comparatively smaller sensible heat flux decrease from the ocean to the atmosphere of 1.47 (0.34 to 2.20) W m$^{-2}$. Over the ocean, latent heat flux from the surface to the
atmosphere is 6.71 (4.95 to 7.89) W m$^{-2}$ lower in G1ocean-albedo than in the preindustrial control simulation. These results indicate a greater shift of energy away from evaporating water and toward increasing land temperature. Large differences in flux magnitude between G1 and G1ocean-albedo can be found over land for net shortwave flux and latent heat flux, and differences in sign can be found over land for total radiative flux. These features are consistent with the applied forcing being different over land and ocean.

## 30   3.5   Hydrological cycle changes

Introducing a strong land-ocean energy and temperature gradient, as in G1ocean-albedo, will undoubtedly impact the hydrological cycle. Although the G1ocean-albedo simulation is idealized, more realistic representations of MCB have shown important hydrological cycle impacts, including secondary circulation patterns that shift precipitation onto land in the tropics and extratropics (Bala et al., 2010; Alterskjær et al., 2013) and changes in the Walker circulation (Niemeier et al., 2013). Here

we evaluate the large-scale hydrological cycle changes in G1ocean-albedo, with possible applicability to other realizations of MCB.

Figure 8 shows global, land, and ocean averaged precipitation, evaporation, and precipitation minus evaporation (P-E) for all of the simulations considered in this manuscript; quantitative descriptions are given in Tables S13-15. The abrupt4xCO2 simulation is the only one with a distinct rapid adjustment and slow response. Over both land and ocean, G1 shows decreases in precipitation and evaporation of approximately equal magnitude, resulting in net changes in P-E of 0.02 (-0.05 to 0.11) mm day$^{-1}$ over land and -0.01 (-0.04 to 0.01) mm day$^{-1}$ over ocean. In G1ocean-albedo, global precipitation and evaporation both decrease by approximately 0.19 (0.11 to 0.26) mm day$^{-1}$ to yield little net change in P-E. However, this net small change is due to differential effects over land and ocean. Over land, precipitation remains relatively unchanged, but evaporation decreases, resulting in a net change in P-E by 0.09 (-0.18 to 0.18) mm day$^{-1}$. Over the ocean, both precipitation and evaporation decrease, with a net negative P-E of -0.06 (-0.19 to -0.01) mm day$^{-1}$.

Annual mean land/ocean contrasts in precipitation and evaporation changes tend to be more uniform in sign in experiment G1 (Figure 9), resulting in few large regions of change in P-E with the exception of the tropics (mostly driven by a southward shift in the intertropical convergence zone; Kravitz et al., 2013a). In G1ocean-albedo, precipitation and evaporation over the oceans are reduced in most regions, consistent with the applied forcing. Over land, the signs of precipitation and evaporation changes are regionally heterogeneous, yet the precipitation and evaporation changes are concordant, e.g., land regions with increased precipitation also generally show increased evaporation. The net P-E map is highly heterogeneous, but in general, tropical land areas are projected to have more available moisture (as measured by P-E) under G1ocean-albedo, and midlatitude land areas are projected to have less. There is a general drying (reduced P-E) in the midlatitudes, as well as some reductions in the intertropical convergence zone, with important implications for tropospheric circulation (to be evaluated in future work).The implications of these changes for people and ecosystems are also important to investigate further.

## 4 Discussion and Conclusions

In Section 3.1, three hypotheses were posed as to why G1ocean-albedo experienced warming over both land and ocean. Energy balance arguments point toward global average warming in G1ocean-albedo. However, energy balance arguments alone cannot explain the magnitude of oceanic warming. Explaining that warming requires a model that can represent horizontal transport of heat from the land to the ocean. Because these processes reach steady state within a decade or less, it is unlikely that long-term oceanic processes, including changes in global mean ocean heat content, are responsible for the majority of the changes seen in G1ocean-albedo.

The results presented here indicate that even though experiments G1 and G1ocean-albedo both achieve approximate net top-of-atmosphere radiative flux balance, the climate system responses differ dramatically between the two experiments. The idea that global energy balance can still result in local changes is perhaps not surprising, as feedbacks operate locally (Armour et al., 2013). These different climate responses for the same magnitude in global forcing are effectively an illustration of different efficacies (Hansen et al., 2005). Even in the absence of slow responses, forcings with different efficacies can cause different

climate system changes (Kravitz et al., 2015). G1ocean-albedo serves as an excellent reminder not to conflate small net top-of-atmosphere radiative flux imbalance with small temperature change; a clear relationship between those two quantities is not guaranteed.

Relatedly, the results obtained for G1ocean-albedo were to some extent by design. The objective of G1ocean-albedo was to achieve net top-of-atmosphere radiative flux balance, which resulted in warming. Conceivably, one could define an objective of no global temperature change, implying a net negative radiative flux at the top-of-atmosphere, or no global land temperature change, requiring adjustments over the oceans to make up the imbalance. It is unclear whether, unlike G1ocean-albedo, such alternate approaches would result in transient behavior that lasts longer than a few years. Such an experiment could be accomplished using feedback methods that have been introduced to geoengineering research in recent years (e.g., MacMartin et al., 2014b; Kravitz et al., 2016).

Related to this discussion, Supplemental Figures S1-S3 show monthly differences (from piControl) of net top-of-atmosphere radiative flux change and temperature change for the abrupt4xCO2, G1, and G1ocean-albedo simulations. These were calculated by naively subtracting each monthly value of the three perturbed experiments from the monthly values of the corresponding piControl simulation, so all differences are subject to noise introduced by chaos. G1 shows an indication of slight transient behavior, starting out with positive temperature anomaly that relaxes to near-zero within a few years. G1ocean-albedo does not show any discernible anomaly, in that it starts out slightly warmer (globally) than piControl and stays slightly warm. The Gregory plot for G1ocean-albedo similarly shows no discernible trend, unlike the abrupt4xCO2 simulation. There are several possibilities of explanations for this behavior. One is that the adjustments are happening on a short enough timescale in G1ocean-albedo that any transient response is difficult to detect with only monthly averages (Cao et al., 2012). Another possibility is that the noise introduced by chaos on the timescales of interest (months to a few years) obscures our ability to detect any transient behavior. An ensemble of shorter simulations (e.g., Wan et al., 2014) might be well equipped to reveal transience in the response on these timescales. A third option is model artifact related to how the climate models treat energy conservation, indicating that experiments like G1ocean-albedo could be useful in testing models beyond their originally conceived application space. While it is beyond the scope of the present manuscript to fully assess all of these possibilities, it becomes clear that G1ocean-albedo and simulations of geoengineering in general are useful for improving understanding about climate modeling and climate science.

The results presented here have several features that were not necessarily expected from the outset. (Kravitz et al., 2013c) found that determining whether the climate system was in balance took up to 30 years of simulation. However, once that balance is achieved, the climate does not change appreciably after the initial rapid adjustment. Potential future work could investigate these results, shedding light on timescales of climate response and potential thresholds, e.g., how large does the energy imbalance need to be to trigger slower adjustments?

Related to this issue of different timescales of adjustment is the traditional separation of climate response into rapid adjustment and slow response components (e.g., Andrews and Forster, 2010; Sherwood et al., 2015). The rapid adjustment is often defined as the climate response unassociated with global mean temperature change, and the slow response describes a transient response due to temperature change, largely as a result of climate system feedbacks. The results from G1ocean-albedo, like

those of G1 (Kravitz et al., 2013b), show an initial rapid change and no appreciable slower change. However, in G1ocean-albedo, there is a sustained temperature increase without appreciable transient behavior. Thus, G1ocean-albedo represents an experiment that does not cleanly delineate into the traditional definitions of rapid adjustment and slow response. Additionally, this sustained temperature increase is to some extent decoupled from net energy imbalances in the climate system, as $\Delta R_{\text{TOA}}$ and $\Delta S$ (Equation 8) are both approximately zero. Reconciling all of these features suggests a potentially rich research topic focused on understanding the relationships between radiative flux changes, temperature changes, and the circumstances under which climate feedbacks are excited, particularly for forcings with strong land/ocean contrast (e.g., anthropogenic aerosols).

The results presented here are broadly relevant to more sophisticated representations of MCB, such as increasing cloud droplet number concentration or directly injecting sea salt aerosols into the marine boundary layer (Kravitz et al., 2013c). Stjern et al. (2018) analyzed a multi-model ensemble of simulations of G4cdnc (Kravitz et al., 2013c), involving a 50% increase in cloud droplet number concentration in all marine low clouds, wherever the model forms those clouds. Although smaller in magnitude, they found similar patterns of top-of-atmosphere radiative flux change as in G1ocean-albedo. Also similar between the two experiments was an increase in land precipitation and a decrease in ocean precipitation. Perhaps an even more realistic representation is G4sea-salt (Ahlm et al., 2017), involving direct injection of sea salt into the marine boundary layer between 30°S and 30°N to achieve an effective radiative forcing of -2.0 W m$^{-2}$. In the injection area (the tropics), this experiment also showed similar patterns of net top-of-atmosphere radiative flux perturbation and hydrologic cycle response. As such, while G1ocean-albedo is highly idealized and exerts a perhaps unrealistically large forcing, it has relevance for other global representations of MCB or sea spray geoengineering. However, there are important differences in boundary layer stability changes from surface albedo increases versus marine cloud brightening. Also, it appears impossible for marine cloud brightening to be conducted over all ocean regions and with a sufficient magnitude to offset the radiative forcing from a quadrupling of the $CO_2$ concentration. The purpose of this manuscript is to describe the broad features of change under a uniform ocean albedo increase, and some of these changes are likely to be present with more realistic scenarios of marine cloud brightening. We anticipate that future research can more deeply explore the applicability of this simulation to marine cloud brightening.

G1ocean-albedo may be more apposite to the impact of geoengineering via "ocean microbubbles," whereby surfactants are added to the ocean surface, promoting the formation of microscopic, highly reflective bubbles (Robock, 2011). An area of investigation we did not undertake, yet one that repeatedly emerges in discussions of microbubbles, is the resulting effects of surface albedo increase on the ocean mixed layer. By reflecting more solar radiation, microbubbles have the potential to inhibit vertical mixing and available light in the euphotic zone, which could have profound effects on marine biota. This implies that another useful future area of investigation for the G1ocean-albedo simulation is an analysis of the marine carbon cycle.

There are numerous potential areas of research prompted by this study. The stark land/ocean contrast in warming has potential implications for ocean dynamics, including the meridional overturning circulation, Western boundary ocean currents, and mixed layer depths, with consequent implications for marine ecosystems and the ocean carbon cycle. This contrast also has implications for the terrestrial biosphere, including ecosystem services and the land and ocean carbon cycles. Although we did not evaluate seasonal changes in this manuscript, such investigations could prove fruitful for more detailed assessments of vari-

ability, such as monsoon precipitation, extreme events, and sea ice extent. In addition, the changes in precipitation described earlier indicate important potential changes in large-scale circulation, atmospheric dynamics, and the hydrological cycle, all of which warrant further study.

*Data availability.* All output involved in the Geoengineering Model Intercomparison Project is publicly available, and much of it is accessible through the Earth System Grid Federation. Please see the GeoMIP website (http://climate.envsci.rutgers.edu/GeoMIP/) or contact the corresponding author for details.

*Competing interests.* None.

*Acknowledgements.* We thank Jón Egill Kristjánsson, who tragically passed away, for invaluable comments on an earlier version of this manuscript. We acknowledge the World Climate Research Programme's Working Group on Coupled Modelling, which is responsible for CMIP, and we thank the climate modeling groups for producing and making available their model output. For CMIP the U.S. Department of Energy's Program for Climate Model Diagnosis and Intercomparison provides coordinating support and led development of software infrastructure in partnership with the Global Organization for Earth System Science Portals. We thank all participants of the Geoengineering Model Intercomparison Project and their model development teams, CLIVAR/WCRP Working Group on Coupled Modeling for endorsing GeoMIP, and the scientists managing the Earth System Grid data nodes who have assisted with making GeoMIP output available. The Pacific Northwest National Laboratory is operated for the U.S. Department of Energy by Battelle Memorial Institute under contract DE-AC05-76RL01830. Simulations performed by Ben Kravitz were supported by the NASA High-End Computing (HEC) Program through the NASA Center for Climate Simulation (NCCS) at Goddard Space Flight Center. Alan Robock is supported by NSF grant AGS-1617844. Olivier Boucher acknowledges HPC resources from CCRT under the allocation 2015-t2012012201 made by GENCI (Grand Equipement National de Calcul Intensif). This work is a contribution to the German DFG-funded Priority Program 'Climate Engineering: Risks, Challenges, Opportunities?' (SPP 1689). Helene Muri was supported by the Research Council of Norway (229760/E10), and acknowledges Sigma2 NOTUR resources nn9182k, NS9033K and nn9448k. U. Niemeier and H. Schmidt are supported by the SPP 1689 within the projects CEIBRAL and CELARIT. This research was supported under the Australian Research Council's Special Research Initiative for the Antarctic Gateway Partnership (project SR140300001). Shingo Watanabe was supported by the Integrated Research Program for Advancing Climate Models, MEXT, Japan. The Earth Simulator was used in his simulations. Jin-Ho Yoon was supported by the National Strategic Project – Find particle of the National Research of Korea (NRF) funded by the Ministry of Science and ICT (MSIT), the Ministry of Environment (ME), and the Ministry of Health and Welfare (MOHW) NRF-2017M3D8A1092022.

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

**Table 1.** Description of the 11 models participating in this study. Column 1 gives the standard model name. Columns 2 and 3 give the default and perturbed surface ocean albedo, defined as upward shortwave divided by downward shortwave radiative flux at the surface, both averaged over ocean regions and over years 11-50 of simulation. Column 4 is the ratio of column 3 to column 2 (calculated prior to rounding the values in Columns 2 and 3). Column 5 gives the factor ($\delta$) by which the model default ocean albedo was multiplied to achieve negligible top-of-atmosphere radiative flux changes under an abrupt4xCO2 simulation (described in greater detail by Kravitz et al., 2015). The differences between Ratio and $\delta$ are caused in part by cloud responses. Column 6 gives a relevant reference for each model. All values are rounded to two decimal places.

| Model name | piControl ocean albedo | G1ocean-albedo ocean albedo | Ratio | $\delta$ | Reference |
| --- | --- | --- | --- | --- | --- |
| BNU-ESM | 0.12 | 0.17 | 1.48 | 2.50 | Ji et al. (2014) |
| CanESM2 | 0.11 | 0.19 | 1.73 | 2.45 | Arora et al. (2011) |
| CESM-CAM5.1-FV | 0.10 | 0.18 | 1.79 | 2.70 | Hurrell et al. (2013) |
| CSIRO-Mk3L-1.2 | 0.12 | 0.19 | 1.61 | 2.04 | Phipps et al. (2011) |
| EC-Earth | 0.10 | 0.19 | 1.97 | 3.17 | Hazeleger et al. (2011) |
| GISS-E2-R | 0.08 | 0.16 | 1.95 | 2.53 | Schmidt et al. (2014) |
| HadGEM2-ES | 0.10 | 0.17 | 1.83 | 2.44 | Collins et al. (2011) |
| IPSL-CM5A-LR | 0.10 | 0.17 | 1.78 | 2.33 | Dufresne et al. (2013) |
| MIROC-ESM | 0.10 | 0.20 | 2.00 | 3.10 | Watanabe et al. (2011) |
| MPI-ESM-LR | 0.09 | 0.23 | 2.40 | 5.42 | Giorgetta et al. (2013) |
| NorESM1-M | 0.09 | 0.18 | 1.95 | 2.77 | Bentsen et al. (2013) |

**Table 2.** Feedback parameters (Section 3.3; units W m$^{-2}$ K$^{-1}$) for global, land, and ocean averages, calculated via the "Gregory method" (Gregory et al., 2004), where annual mean top-of-atmosphere net radiative flux is regressed against annual mean temperature.

|  | Global feedback parameter ($\lambda_g$) | Land feedback parameter $\lambda_\ell$ | Ocean feedback parameter $\lambda_o$ |
| --- | --- | --- | --- |
| BNU-ESM | 0.9019 | 0.7181 | 0.9838 |
| CanESM2 | 1.1539 | 1.1898 | 1.1260 |
| CESM-CAM5.1-FV | 1.1435 | 1.0357 | 1.1591 |
| CSIRO-Mk3L-1.2 | 1.0192 | 0.9300 | 0.8034 |
| EC-Earth | 1.2124 | 1.1937 | 1.3155 |
| GISS-E2-R | 2.2440 | 1.9751 | 2.3560 |
| HadGEM2-ES | 0.8411 | 0.8363 | 0.8351 |
| IPSL-CM5A-LR | 0.8367 | 1.2891 | 0.5894 |
| MIROC-ESM | 1.0378 | 0.8736 | 1.0383 |
| MPI-ESM-LR | 1.3701 | 1.0573 | 1.3986 |
| NorESM1-M | 1.4285 | 1.8828 | 1.6063 |

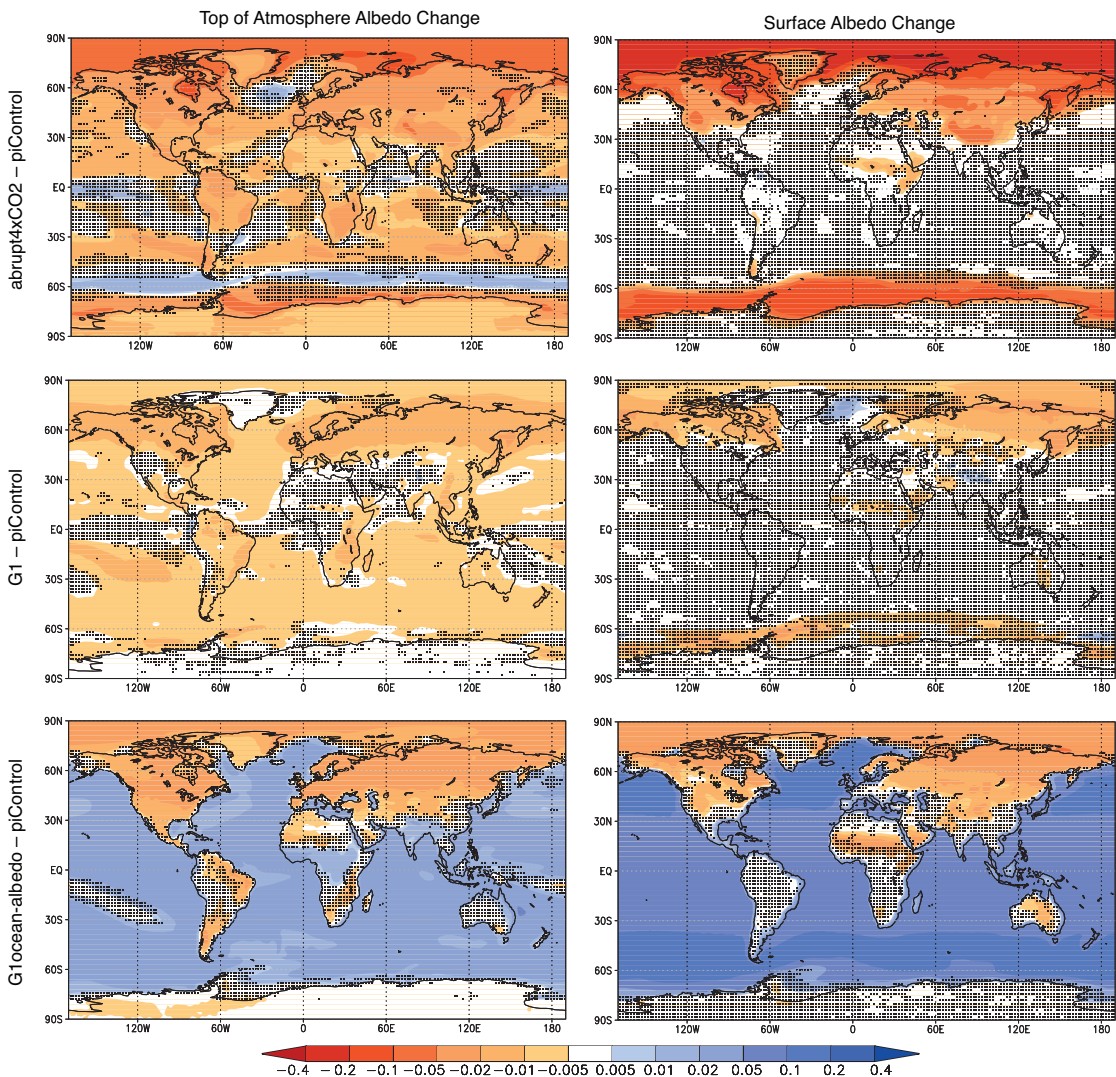

**Figure 1.** Top-of-atmosphere (TOA) and surface albedo differences (relative to piControl) for the abrupt4xCO2, G1, and G1ocean-albedo experiments. Albedo here is calculated as the ratio of upwelling to downwelling all-sky shortwave radiative flux, either at TOA or at the surface. Values are averages over years 11-50 of simulation. Stippling indicates where fewer than 8 out of 11 models agree on the sign of the response.

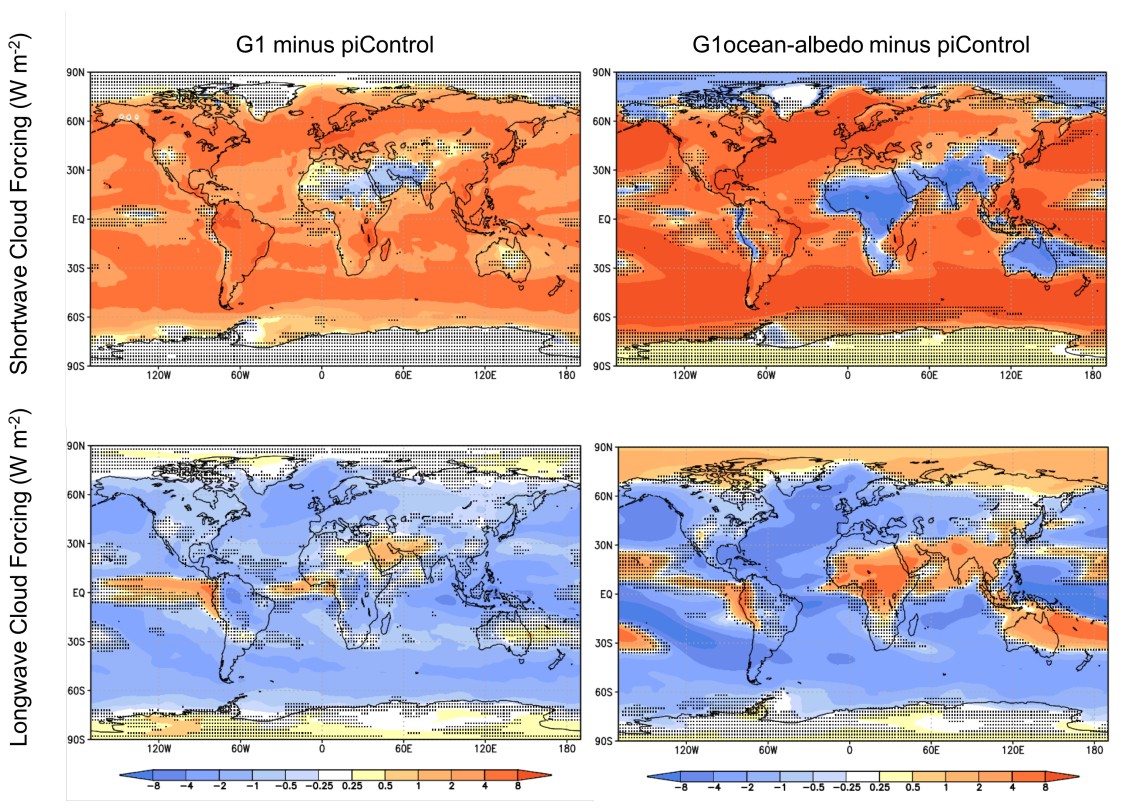

**Figure 2.** Shortwave (top) and longwave (bottom) cloud forcing changes due to the G1 (left) and G1ocean-albedo perturbations. Cloud forcing is defined as all-sky minus clear-sky radiative flux at the top of the atmosphere, with positive values indicating more net downward flux.

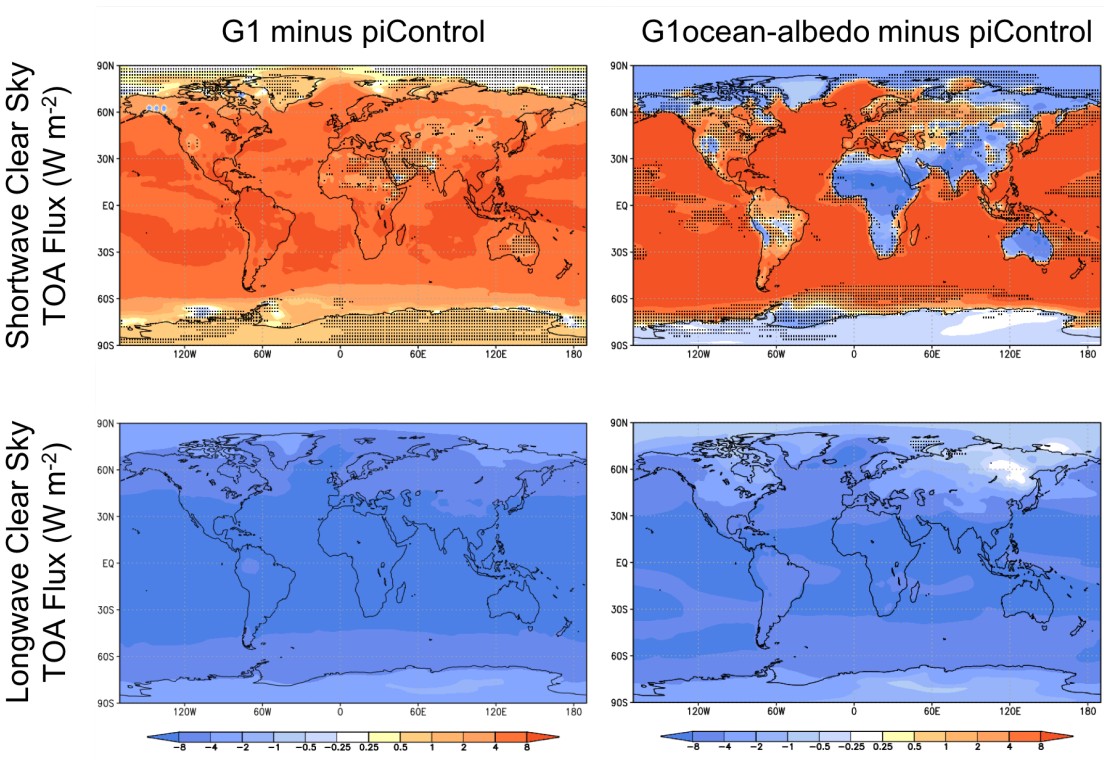

**Figure 3.** Shortwave (top) and longwave (bottom) net (downward minus upward) clear sky radiative flux changes at the top-of-atmosphere due to the G1 (left) and G1ocean-albedo perturbations, with positive values indicating more net downward flux. Positive values indicate that upward clear sky flux decreased in the perturbed (G1 or G1ocean-albedo) experiments, and negative values indicate that upward clear sky flux increased in the perturbed experiments.

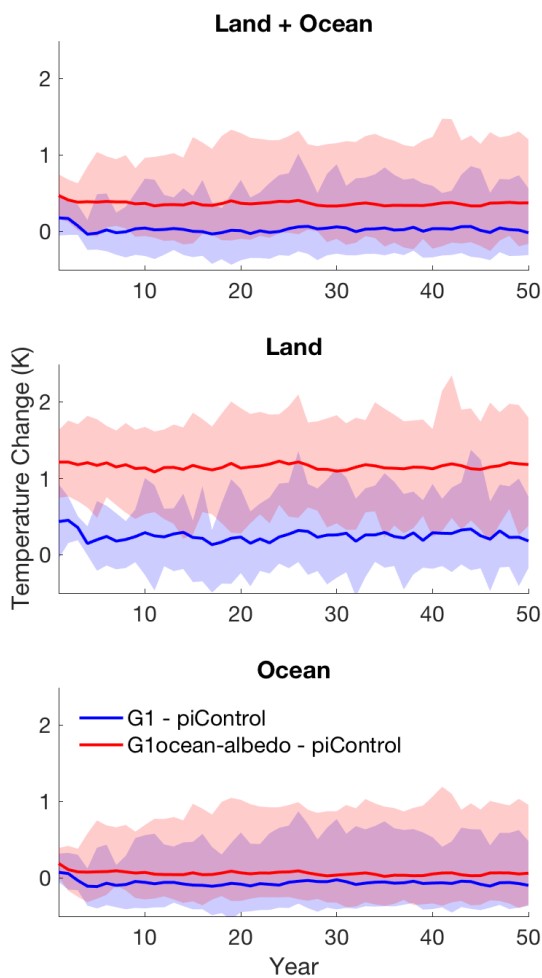

**Figure 4.** Global (top), land (middle), and ocean (bottom) average temperature change for the G1 (blue) and G1ocean-albedo (red) simulations. Lines show the all-model ensemble mean, and shading shows model spread (smallest to largest values).

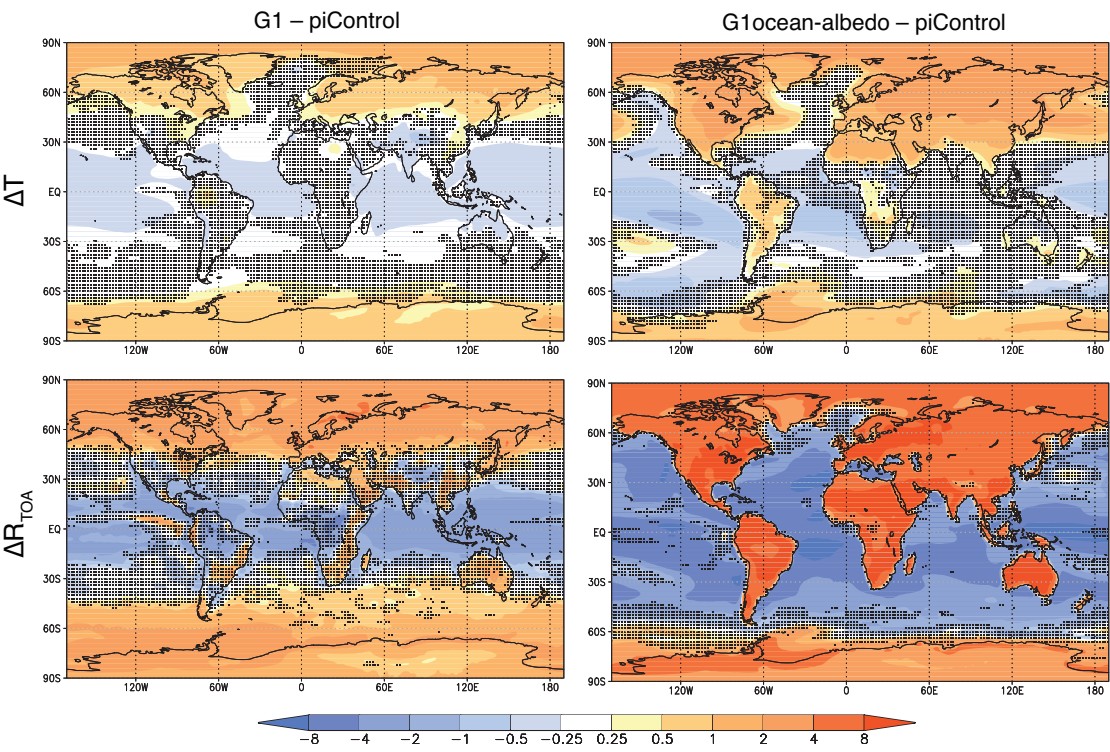

**Figure 5.** Surface air temperature (top row; K) and TOA net radiative flux (bottom row; W m$^{-2}$) changes for experiments G1 (left) and G1ocean-albedo (right). Values are averages over years 11-50 of simulation. Stippling indicates where fewer than 8 out of 11 models agree on the sign of the response.

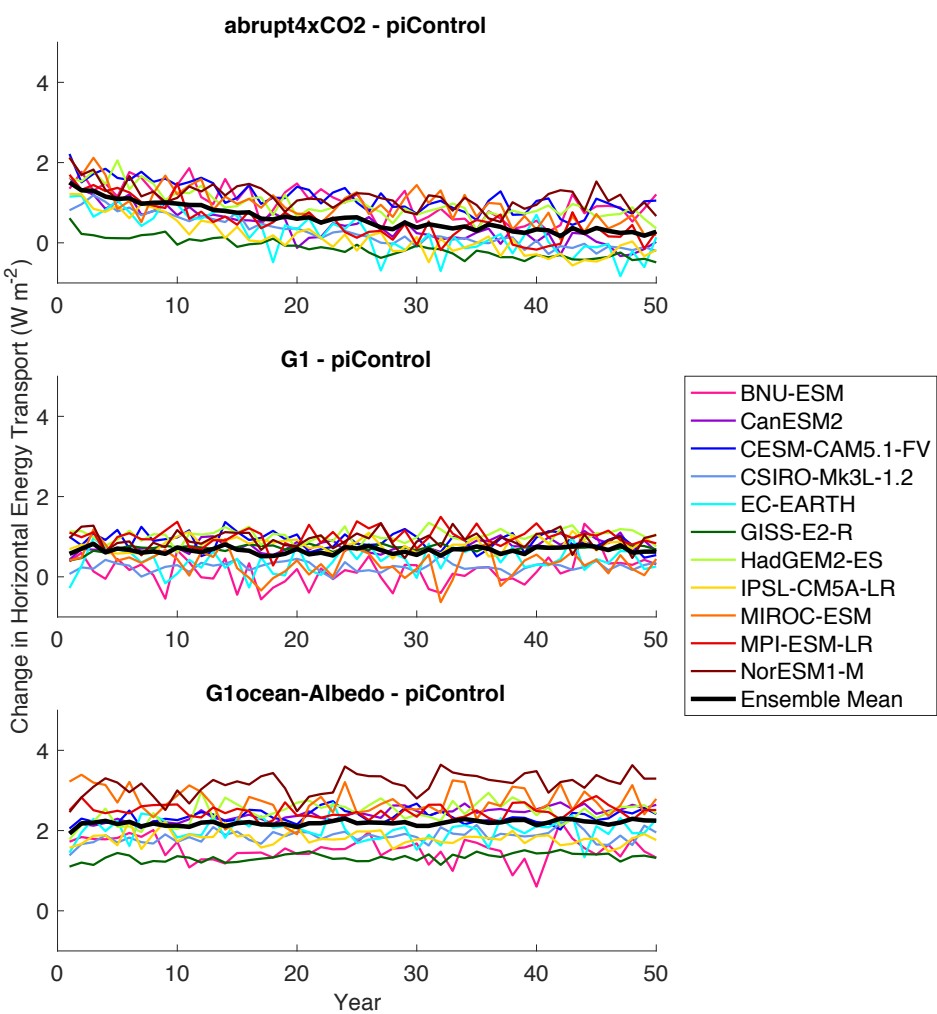

**Figure 6.** Annual mean change in land-ocean energy transport (Section 3.3; W m$^{-2}$) from piControl. See Equation 7 for a formal definition.

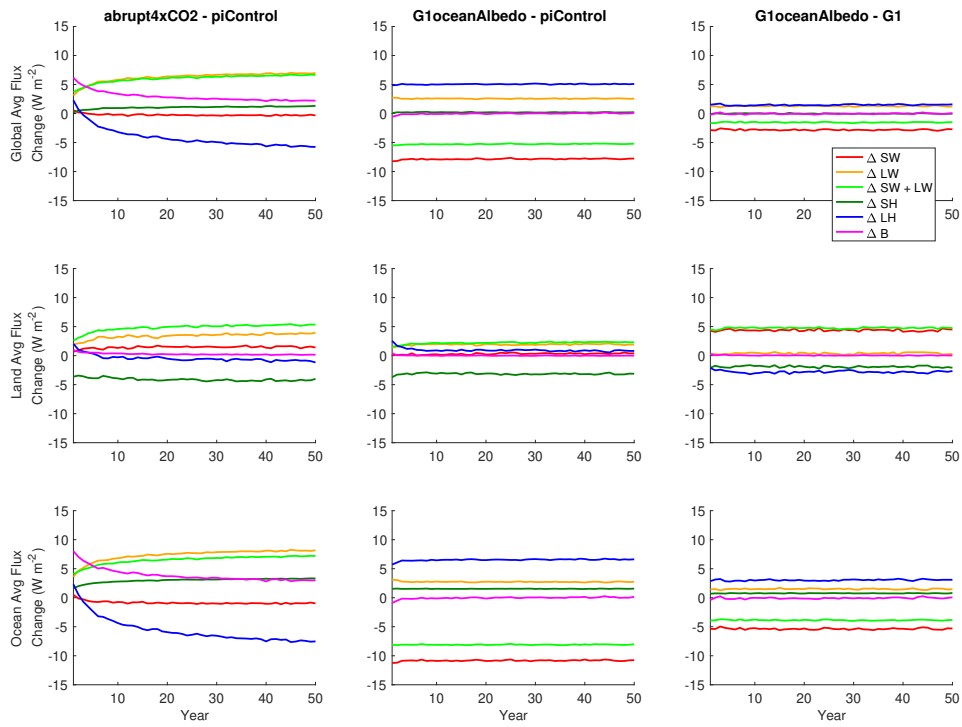

**Figure 7.** Annual mean time series of all-model mean surface fluxes (terms in Equation 8) for global averages (top), land averages (middle), and ocean averages (bottom). All fluxes are positive in the downward direction.

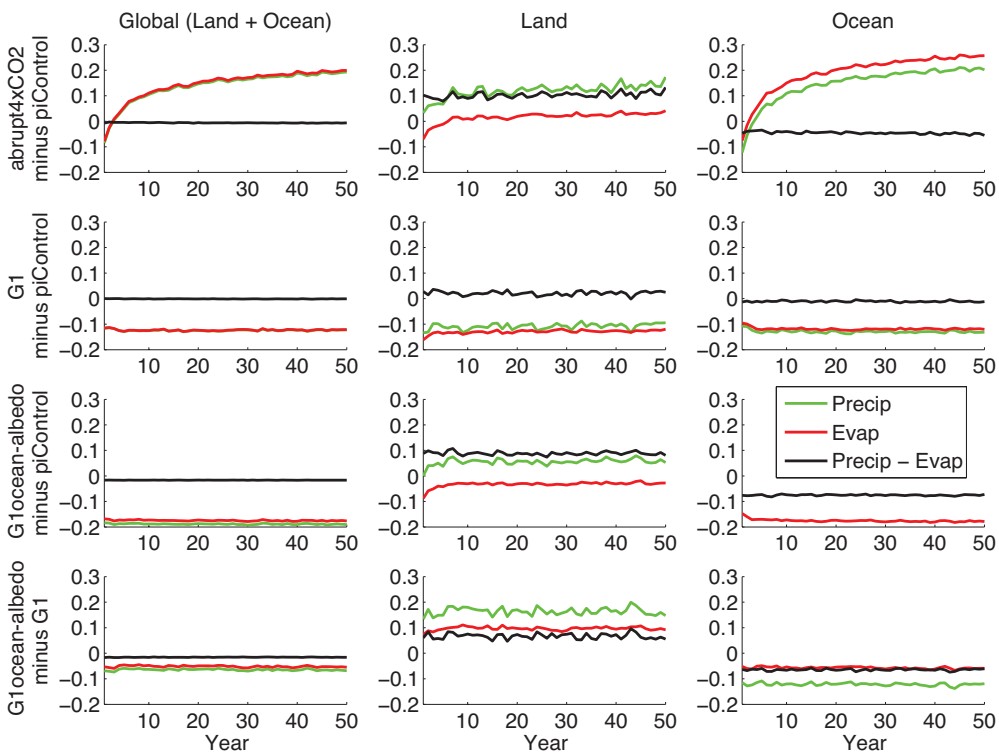

**Figure 8.** Annual mean time series of hydrological cycle changes (all in mm day$^{-1}$). Green lines show precipitation changes, red lines show evaporation changes, and black lines show precipitation minus evaporation. In the first column, green lines are difficult to see because they are largely overlaid by red lines. In the third row, third column, the green line has values below -0.2 for all years.

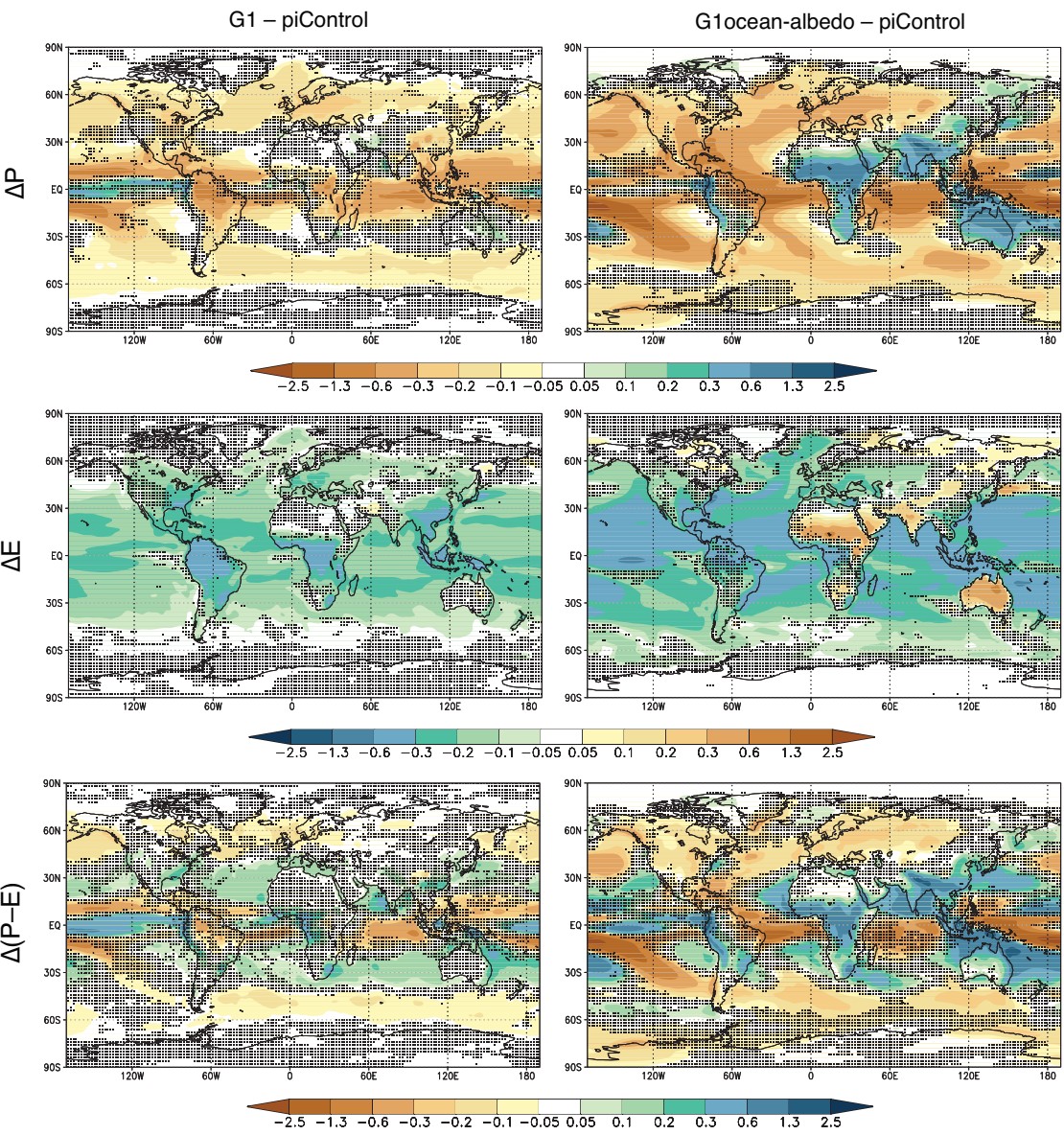

**Figure 9.** Precipitation (top row), evaporation (middle row), and precipitation minus evaporation (bottom row) changes (all panels have units mm day$^{-1}$) for experiments G1 and G1ocean-albedo. Values are averages over years 11-50 of simulation. Stippling indicates where fewer than 8 out of 11 models agree on the sign of the response.