# Peer review of "The climate effects of increasing ocean albedo: An idealized representation of solar geoengineering"

_Atmospheric Chemistry and Physics, 2018_

## Short Comment (SC1) · 2 May 2018

R. Seitz
russellseitz@gmail.com

I am surprised that in discussing an experiment that finds:

> Eleven Earth System Models are relatively consistent in their temperature, radiative flux, and hydrological cycle responses...

GeoMIP should elide:

> an abrupt quadrupling of ocean albedo to maintain approximate net top-of-atmosphere radiative flux balance "

with the doubling of $CO_2$ in one of the references it discusses. Seitz 2011 reported the initial CAM3.1 modeling of the combined effect of higher ocean albedo and $CO_2$ doubling to 780ppm, approximating the forcing in IPCC Representative Concentration Pathway 6.0

In contrast, Kravitz et al 2018 instantaneously quadruple $CO_2$ to 1600 ppm, considerably outside the IPCC envelope, which only extends to ~1230 ppm $CO_2$ eq. in RCP 8.5, as seen in the accompanying graph.

[Figure]

1600 * Kravitz et al 2018

Concentration - CO$_2$-eq. (incl. all forcing agents)

780 * Seitz 2011

CO$_2$-eq. (ppm)

← MESSAGE - RCP 8.5   ■ AIM - RCP 6.0   ▲ MiniCAM - RCP 4.5   ● IMAGE - RCP3-PD (2.6)

While Seitz 2011 is primarily concerned with fresh water conservation, it used the CAM3.1 model to quantify the coupled climate impact of increased sea surface albedo and doubled CO2 forcing ,and found substantial continental cooling in such a global case, using  a carbon forcing well within the IPCC parameter envelope-  780 PPM C02 eq. , which approximates the  RCP 6.0 projection for 2100.

In contrast, *The climate effects of increasing ocean albedo: An idealized representation of solar geoengineering* considers a more dystopic future.

As the authors candidly note:

> "The results obtained for G1ocean-albedo were to some extent by design. The objective of G1ocean-albedo was to achieve net top-of-atmosphere radiative flux balance, which resulted in warming.
> Conceivably, one could define an objective of no global temperature change, implying a net negative radiative flux at the top-of-atmosphere, or no global land temperature change, requiring adjustments over the oceans to make up the imbalance. It is unclear whether, unlike G1ocean-albedo, such alternate approaches would result in transient behavior that lasts longer than a few years. Such an experiment could be accomplished using feedback methods that have been introduced to geoengineering research in recent years (e.g., MacMartin et al., 2014b; Kravitz et al., 2016)."

It is natural to assume that model intercomparison experiments involve models with comparable forcings, use reasonably physical feedback parametrizations, and rely on the objective description of cited works. This is not evident in the following passage:

> "G1 ocean-albedo may be more apposite to the impact of geoengineering via "ocean microbubbles," whereby surfactants are added to the ocean surface, promoting the formation of microscopic, highly reflective bubbles (Seitz, 2011; Robock, 2011).

This does not represent the content of Seitz 2011, which speaks for itself -- neither its title: *Bright Water: Microbubbles, water conservation and climate change*, or its text contains the word 'geoengineering.'

It discusses the physics of reducing solar forcing in the hydrosphere, not the atmosphere, and while discussing the

relationship between microbubble lifetime and variable ocean biochemistry that gives rise to natural albedo effects like undershine, does not presume adding surfactants to promote their formation- none were used in the (physical) experiments in albedo modification it describes.

The authors continue:

> An area of investigation we did not undertake, yet one that repeatedly emerges in discussions of microbubbles is the resulting effects of surface albedo increase on the ocean mixed layer. By reflecting more solar radiation, microbubbles have the potential to inhibit vertical mixing and available light in the euphotic zone, which could have profound effects on marine biota. This implies that another useful future area of investigation for the G1 ocean-albedo simulation is an analysis of the marine carbon cycle.

This reflects Robock 2011, an Editorial Comment in *Climatic Change* entitled 'Bubble Bubble Toil and Trouble' and these issues are addressed at some length in in Seitz 2011, which called for their investigation by ecologists and systems biologists. Readers should also note that as modeled with CAM3.1, water brightening reduced peak water surface temperature, which could promote, rather than inhibit, diurnal mixing, by reducing thermal stratification.

I appreciate the utility of idealized simulations, but fear this article invites something best avoided—policy analysts may confuse idealizations with models of the real world. It would clearly be desirable to apply approximations of water reflectivity and temperature less naive that simple Mie theory, or unmixed slab models, to the complex evaporative response of both fresh water reservoirs and the sea surface, with a view to achieving less

idealized and more realistic representations of solar radiation management, especially on local scales.

Once again my thanks to Ken Caldeira for the CAM-1 modeling that informed Seitz 2011.

---

## Referee Comment (RC1) · S. Tilmes (Referee) · 25 May 2018

The paper discusses climate effects of the ocean albedo GeoMIP experiments. It is well written and organized. I support the publication in ACP after the following minor comments have been addressed.

Minor Comments

Page 2, Line 11: "stratospheric sulfate aerosol geoengineering"; please consider adding an abbreviation here, to refer to later in the text.

Figure 1 is showing the albedo change. Why is the color scale (red negative) different

from Figures 2 and 3 (where red indicates a positive change)?

Figure 2 and 3, please make sure to add the direction of the forcings shown in the figure caption (positive downward or upward), which will help explain these figures in the text (e.g., Page 4, line 30: so it makes sense why positive shortwave values indicate a reduction in clouds).

Page 8, Equation 8: It would be helpful to describe in more detail how this equation is coming about, either a reference or a more detailed derivation.

Equation 9: Delta A is defined as the time-dependent LOET (W/m2). However, it was also defined earlier as albedo changes, please change.

Line 17: In Experiment -> In experiment Line 25: "necessarily" does not make sense, please remove. Line 30ff: These lines does not make much sense, since it seems that the calculations are not right, why discuss them, and why not just remove Line 25-33?

Figure 7 caption: change Equation 3 to Equation 10.

Equation 10: Delta S is the net surface flux, but earlier it is defined as solar radiation. It is confusion to use the same acronyms for different quantities.

Page 10, Line 6: What do you mean by land processes? If you see a reduction in SH (so warming) over land and an increase in LH (so reduction in evaporation), you could say that "... are due to changes in the turbulent fluxes over land rather than ocean heat content reductions, and those also increase temperatures over the ocean through LOET." However, I am not convinced that the discussion on surface energy fluxes justifies any conclusions on ocean heat uptake changes. One way to look at this is actually looking at quantities like ocean temperatures, AMOC changes. Also, changes over the ocean seem to be strongly latitudinal dependent. The continued warming of high latitudes likely results in continued sea ice reduction and potentially decrease salinity, which may further slowdown the AMOC and reduce heat update in the ocean. Changes in precipitations would also play a role in changing runoff and

salinity. Therefore, I think, from the global investigations given in this paper, one has to be careful to draw any conclusions.

Page 11, Line 5: G1ocean-albedo seems to show a drying band over mid-latitudes between 30-60 degrees and a potential reduction over the ITCZ, which can be pointed out in the paper. Changes in ocean albedo therefore could have specific impacts on tropospheric circulation. Please comment.

Page 11: 11-16: The first 6 lines of the conclusions seem to give the same message in different ways, this could be shortened.

Discussion and conclusions: The authors discuss various topics that need to be still investigated, including different experimental design, timescales for climate response, questions on fast and slow response, going away from the science presented in the current study (over 2 pages).

Instead of discussing the various research areas somewhat related to this topic, I suggest to shorten this part substantially and would rather appreciate a discussion and summary closer to the findings of this paper. What was the result of investigating the 3 hypotheses? There seems to be still a lot of open questions on the impacts of G1ocean-albedo that are directly related to the results in this paper that could be mentioned, for instance more research is needed to understand the changes in atmospheric dynamics and temperatures (beyond surface temperatures), hydrological cycle and ocean changes, as partly discussed in the last section.

Figure 3S, add captions for experiments shown in each panel
* * *

---

## Referee Comment (RC2) · Anonymous Referee #2 · 5 Jun 2018

The paper discusses the impacts of spatially heterogeneous, near-surface solar geo-engineering techniques within the idealised scenario of a homogeneous positive scaling in surface ocean albedo. The results are contrasted with solar geoengineering techniques impacting solar irradiance at the top of atmosphere. The multi-model simulations were performed within the GeoMIP initiative. The paper is well conceived and well written. The results are of high relevance to the community and I recommend publication following only minor adjustments of the manuscript.

Minor Comments:

P1L2: I would suggest to view these experiments as an idealized representation of

near-surface solar geoengineering approaches over the oceans in general. I personally would also draw a stronger connection to ocean surface albedo techniques rather than MCB. While the comparison is justified (both being geoengineering techniques associated with a regionally heterogeneous forcing), I remain unconvinced that G1ocean-albedo constitutes "an idealized representation of marine cloud brightening". Ocean albedo modifications are most effective in the clear-sky and if anything reduce the radiative impact of clouds on the SW surface balance (contrarily to MCB). Therefore the analogue seems hard to justify physically in my opinion.

P5L23ff & Fig.5 : The persistent warming of surface air temperature to the east of the continents in the Northern Hemisphere is interesting. Is there any seasonality associated with this warming? i.e. winter or summer?

P6L20: I remain unconvinced by the assumption dA=0 for G1. Equation 5 does not seem essential and equation 6 would merely contain another term, which you can estimate?

Page 6 L20ff: I suggest to restructure the discussion of this section. It is sometimes difficult to follow which part of the discussion is applicable to G1 and which to G1ocean-albedo. It may be worth to split the discussion going through the assumptions of G1 and G1ocean-albedo separately and followed by a discussion contrasting these two. A few concrete suggestions follow:

L21: Rephrase "of this equation is equal to" as "of Equ. 5 equals"

P7L11ff: The paragraph starts with the discussion on assuming dA=0 for the G1 simulations and ends with the change in dTs,o for G1ocean-albedo. It is not clear to me, why one would use the $d\varepsilon$ from G1 to estimate dTs,o for G1ocean-albedo.

Equ. 9: A is already used for albedo.
* * *

---

## Author Comment (AC1) · 30 Jul 2018

We have uploaded a response to all reviewers in a single document. Following the response is a tracked changes version of the manuscript.

Please also note the supplement to this comment: https://www.atmos-chem-phys-discuss.net/acp-2018-340/acp-2018-340-AC1-supplement.pdf

---

## Author Response (AR1)

Response to reviewers
Kravitz et al.
acp-2018-340

Original reviewer comments in plain font.  **Responses in bold.**

Please also see attached a "tracked changes" version of the manuscript incorporating all of our revisions.
* * *
*Reviewer #1 (Simone Tilmes)*

The paper discusses climate effects of the ocean albedo GeoMIP experiments. It is well written and organized. I support the publication in ACP after the following minor comments have been addressed.

**Thank you!**

Minor Comments

Page 2, Line 11: "stratospheric sulfate aerosol geoengineering"; please consider adding an abbreviation here, to refer to later in the text.

**We struggled with finding a satisfactory acronym.  Given that this paper is not directly about stratospheric sulfate aerosol geoengineering, we decided to omit the acronym and spell out the entire term the few times we use it.**

Figure 1 is showing the albedo change. Why is the color scale (red negative) different from Figures 2 and 3 (where red indicates a positive change)?

**We felt that the color scale was more intuitive this way.  Figure 1 shows albedo changes, where blue is a higher albedo (less net influx of radiation), and red is a lower albedo (more net influx of radiation).  In Figures 2 and 3, we use the same scheme – red means more net influx of radiation, and blue means less net influx of radiation.**

Figure 2 and 3, please make sure to add the direction of the forcings shown in the figure caption (positive downward or upward), which will help explain these figures in the text (e.g., Page 4, line 30: so it makes sense why positive shortwave values indicate a reduction in clouds).

**We agree.  Added.**

Page 8, Equation 8: It would be helpful to describe in more detail how this equation is coming about, either a reference or a more detailed derivation.

**We agree.  We neglected to include a citation here, which we have now added.**

Equation 9: Delta A is defined as the time-dependent LOET (W/m2). However, it was also defined earlier as albedo changes, please change.

**Thanks for catching that. Fixed.**

Line 17: In Experiment -> In experiment Line 25: "necessarily" does not make sense, please remove. Line 30ff: These lines does not make much sense, since it seems that the calculations are not right, why discuss them, and why not just remove Line 25-33?

**We agree and have removed lines 25-33.**

Figure 7 caption: change Equation 3 to Equation 10.

**Changed. Thanks for catching that.**

Equation 10: Delta S is the net surface flux, but earlier it is defined as solar radiation. It is confusion to use the same acronyms for different quantities.

**Oops. Changed.**

Page 10, Line 6: What do you mean by land processes? If you see a reduction in SH (so warming) over land and an increase in LH (so reduction in evaporation), you could say that "... are due to changes in the turbulent fluxes over land rather than ocean heat content reductions, and those also increase temperatures over the ocean through LOET." However, I am not convinced that the discussion on surface energy fluxes justifies any conclusions on ocean heat uptake changes. One way to look at this is actually looking at quantities like ocean temperatures, AMOC changes. Also, changes over the ocean seem to be strongly latitudinal dependent. The continued warming of high latitudes likely results in continued sea ice reduction and potentially decrease salinity, which may further slowdown the AMOC and reduce heat update in the ocean. Changes in precipitations would also play a role in changing runoff and salinity. Therefore, I think, from the global investigations given in this paper, one has to be careful to draw any conclusions.

**There are multiple points made in this comment, so we address each one separately:**
- **We remain agnostic about land processes, because we did not do a thorough diagnosis as to which processes might be responsible for the changes. The main thrust of our argument is that the timescale of response is consistent with land processes and feedbacks. We have updated the language to better describe what we mean.**
- **We are unsure how to modify the paper in response to the reviewer's second point, in that over land in G1ocean-albedo, changes in both SH and LH are negative (Figure 7). Moreover, the changes in individual heat fluxes that the reviewer mentions are diagnostic, and they do not reveal causality.**

- **We agree that our discussion of ocean heat uptake was insufficient. We had intended for the discussion to be solely about mean changes, and we acknowledge that there may be a role for regional changes or modes of circulation in altering ocean heat uptake. We have adjusted the language in the paper to soften our statements and to specify when we are discussing mean changes. We have also added a mention that regional changes may be important. Nevertheless, we feel as though the argument about timescales is relevant, and we are not convinced that any of the mentioned processes (e.g., AMOC) can operate on a short enough timescale with sufficient strength to explain the changes observed.**

Page 11, Line 5: G1ocean-albedo seems to show a drying band over mid-latitudes between 30-60 degrees and a potential reduction over the ITCZ, which can be pointed out in the paper. Changes in ocean albedo therefore could have specific impacts on tropospheric circulation. Please comment.

**We agree with this point. We have now mentioned the importance of these features in the paper and indicated that they could imply changes in tropospheric circulation. So as not to provide too many messages in a single paper, we reserve analyses of circulation for future work.**

Page 11: 11-16: The first 6 lines of the conclusions seem to give the same message in different ways, this could be shortened.

**We evaluated these sentences, and while the differences between the sentences may be subtle, we feel that they are arguing slightly different, important points. As such, we are worried that making cuts would remove some important messages. If the reviewer has specific suggestions to this end, we would be happy to consider them.**

Discussion and conclusions: The authors discuss various topics that need to be still investigated, including different experimental design, timescales for climate response, questions on fast and slow response, going away from the science presented in the current study (over 2 pages). Instead of discussing the various research areas somewhat related to this topic, I suggest to shorten this part substantially and would rather appreciate a discussion and summary closer to the findings of this paper. What was the result of investigating the 3 hypotheses? There seems to be still a lot of open questions on the impacts of G1ocean-albedo that are directly related to the results in this paper that could be mentioned, for instance more research is needed to understand the changes in atmospheric dynamics and temperatures (beyond surface temperatures), hydrological cycle and ocean changes, as partly discussed in the last section.

**We agree that we neglected to include a summary of our results, which we feel would be valuable; we have added a paragraph in the final section. We also agree that we could have included more discussion of these future areas of work, which we have now added.**

**We are reluctant to remove or shorten the discussions contained in this section, as we believe they are germane to the study.  For example, the discussion of fast and slow responses is essential, as through that discussion, new discoveries are revealed:  (1) the experiment stays in balance after a few years, indicating little slow response, which is unlike what was found in the initial paper discussing the experiment design; (2) this lack of slow response is in the presence of global mean temperature change, which is somewhat counter to the definition of slow response, indicating that G1ocean-albedo can provide new insight about the radiative forcing and climate response framework.  Some of the other discussion includes how this study is relevant to marine cloud brightening or near-surface ocean geoengineering, which we feel is important to discuss.  Those two areas alone take up about 2 pages of text, and we feel that making this section shorter would lose some of those important points.**

Figure 3S, add captions for experiments shown in each panel.

**Added.**
* * *
*Reviewer #2 (Anonymous)*

The paper discusses the impacts of spatially heterogeneous, near-surface solar geoengineering techniques within the idealised scenario of a homogeneous positive scaling in surface ocean albedo. The results are contrasted with solar geoengineering techniques impacting solar irradiance at the top of atmosphere. The multi-model simulations were performed within the GeoMIP initiative. The paper is well conceived and well written. The results are of high relevance to the community and I recommend publication following only minor adjustments of the manuscript.

**Thank you!**

Minor Comments:

P1L2: I would suggest to view these experiments as an idealized representation of near-surface solar geoengineering approaches over the oceans in general. I personally would also draw a stronger connection to ocean surface albedo techniques rather than MCB. While the comparison is justified (both being geoengineering techniques associated with a regionally heterogeneous forcing), I remain unconvinced that G1ocean- albedo constitutes "an idealized representation of marine cloud brightening". Ocean albedo modifications are most effective in the clear-sky and if anything reduce the radiative impact of clouds on the SW surface balance (contrarily to MCB). Therefore the analogue seems hard to justify physically in my opinion.

**This is a valid point.  We have altered the language in the abstract, introduction, and conclusions to better take this distinction into account.**

P5L23ff & Fig.5 : The persistent warming of surface air temperature to the east of the continents in the Northern Hemisphere is interesting. Is there any seasonality associated with this warming? i.e. winter or summer?

**Not really. We looked at individual months in individual models (not shown), and while some models show a stronger response than others, that signature of warming is visible pretty much all the time in every model. We have added a sentence to this effect.**

P6L20: I remain unconvinced by the assumption dA=0 for G1. Equation 5 does not seem essential and equation 6 would merely contain another term, which you can estimate?

**You're right. This was not an appropriate assumption and not really necessary. We have revised this section accordingly.**

Page 6 L20ff: I suggest to restructure the discussion of this section. It is sometimes difficult to follow which part of the discussion is applicable to G1 and which to G1ocean- albedo. It may be worth to split the discussion going through the assumptions of G1 and G1ocean-albedo separately and followed by a discussion contrasting these two. A few concrete suggestions follow:

**We thank the reviewer for this suggestion and have reorganized this entire section accordingly.**

L21: Rephrase "of this equation is equal to" as "of Equ. 5 equals"

**After the reorganization, this sentence no longer exists.**

P7L11ff: The paragraph starts with the discussion on assuming dA=0 for the G1 simulations and ends with the change in dTs,o for G1ocean-albedo. It is not clear to me, why one would use the dε from G1 to estimate dTs,o for G1ocean-albedo.

**You're right, the assumption of dA=0 was not appropriate to do. We've gone through the section and further justified our choices. We acknowledge that we still use dε from G1 to estimate dTs,o for G1ocean-albedo. Without doing so, we were unable to come up with a way of obtaining estimates of dTs,o for G1ocean-albedo. We discuss this in more detail in the paper and justify our choice, as well as provide caveats and estimates as to how this choice may have affected our results.**

Equ. 9: A is already used for albedo.

**Thanks for catching that. Fixed.**
* * *
*Short Comment #1 (Russell Seitz)*

We thank Dr. Seitz for his comment and apologize for our incorrect citation of his paper, which we agree misrepresented his ideas. We appreciate his suggestions for alternate citations and have used those instead.

[revised manuscript text omitted]
. ~~Previous studies suggest that the clouds that are most susceptible to albedo modification are located on the eastern side of ocean basins in the subtropics, in regions dominated by stratocumulus clouds (Oreopoulos and Platnick, 2008). So, rather than considering a reduction of solar input operating uniformly over both land and ocean, an idealized representation of MCB that better approximates the effects is to increase the albedo only over ocean surfaces, as described by Kravitz et al. (2013c). This method can also be used to assess some effects of geoengineering by creating microbubbles at the ocean surface to increase reflectivity (e.g., Seitz, 2011; Robock, 2011; Gabriel et al., 2017).~~

[revised manuscript text omitted]

$$\frac{d\epsilon/2}{1-\epsilon/2} = \frac{-dS(1-A)}{4(1-\epsilon/2)\sigma T_s^4}$$

5    ~~In Equation 4 , the first term is for solar changes, the second term is for planetary albedo changes, and the third term is for emissivity (greenhouse gas) changes. The emissivity term yields a temperature change of approximately 3.01 K. The albedo change term (for G1ocean-albedo only) with $dA = 0.0145$ (Supplemental Table S3) yields a temperature change of -1.48 K. The solar reduction term (for G1 only) yields a temperature change of -3.01 K. As such, the net temperature change for G1 is by construction approximately 0, and the net temperature change for G1ocean-albedo would be $dT_s = 1.53$. As such, under~~

[revised manuscript text omitted]

Experiment G1ocean-albedo exhibits a strong land-ocean contrast in temperature (Figure 5), and the response is in steady

20   state after a few years. As such, consistent with the behavior of other fluxes, LOET in G1ocean-albedo does not show transient behavior. LOET in G1ocean-albedo is approximately 2.20 (1.35 to 3.21) W m$^{-2}$, which is larger than in the other experiments examined here.

25   ~~specifically, the temperature "added" to the air over the oceans by LOET can be calculated as $\Delta A/\lambda_o$, and the temperature "subtracted" from the air over land by LOET can be calculated as $\Delta A/\lambda_\ell$. Performing these calculations, LOET in the G1ocean-albedo experiment contributes 1.87 K (0.57 to 3.06) to ocean temperature and "subtracts" 2.03 K (0.68 to 3.06) from land temperature. We caution that this naive calculation is somewhat circular, and it inherently includes in the ocean calculations both regions unaffected by LOET (e.g., tropical oceans) and regions strongly affected by LOET (e.g., Northeast~~

[revised manuscript text omitted]

10 Schmidt, G. A., Kelley, M., Nazarenko, L., Ruedy, R., Russell, G. L., Aleinov, I., Bauer, M., Bauer, S. E., Bhat, M. K., Bleck, R., Canuto, V., Chen, Y.-H., Cheng, Y., Clune, T. L., Genio, A. D., de Fainchtein, R., Faluvegi, G., Hansen, J. E., Healy, R. J., Kiang, N. Y., Koch, D., Lacis, A. A., LeGrande, A. N., Lerner, J., Lo, K. K., Matthews, E. E., Menon, S., Miller, R. L., Oinas, V., Oloso, A. O., Perlwitz, J. P.,

Puma, M. J., Putman, W. M., Rind, D., Romanou, A., Sato, M., Shindell, D. T., Sun, S., Syed, R. A., Tausnev, N., Tsigaridis, K., Under, N., Volugarakis, A., Yao, M.-S., and Zhang, J.: Configuration and assessment of the GISS ModelE2 contributions to the CMIP5 archive, J. Adv. Modell. Earth Syst., 6, 141–184, https://doi.org/10.1002/2013MS000265, 2014.

Schmidt, H., Alterskjær, K., Karam, D. B., Boucher, O., Jones, A., Kristjánsson, J. E., Niemeier, U., Schulz, M., Aaheim, A., Benduhn, F., Lawrence, M., and Timmreck, C.: Solar irradiance reduction to counteract radiative forcing from a quadrupling of $CO_2$: Climate responses simulated by four Earth system models, Earth System Dynamics, 3, 63–78, https://doi.org/10.5194/esd-3-63-2012, 2012.

Seitz, R.: Bright water: hydrosols, water conservation and climate change, Climatic Change, 105, 365–381, https://doi.org/10.1007/s10584-010-9965-8, 2011.

[revised manuscript text omitted]